# A measure of centrality in cyclic diffusion processes: Walk-betweenness

**Yoosik Youm** [1]\*, **Byungkyu Lee** [2], **Junsol Kim** [1]

1 Sociology Department, Yonsei University, Seoul, South Korea, 2 Sociology Department, Indiana University, Bloomington, IN, United States of America

\* yoosik@yonsei.ac.kr

## Abstract

Unlike many traditional measures of centrality based on *paths* that do not allow any repeated nodes or lines, we propose a new measure of centrality based on *walks*, *walk-betweenness*, that allows any number of repeated nodes or lines. To illustrate the value of walk-betweenness, we examine the transmission of syphilis in Chicago area and the diffusion of microfinance in 43 rural Indian villages. Walk-betweenness allows us to identify hidden bridging communities in Chicago that were essential in the transmission dynamics. We also find that village leaders with high walk-betweenness are more likely to accelerate the rate of microfinance take-up among their followers, outperforming other traditional centrality measures in regression analyses.

## Introduction

We propose a new bridging (brokerage) centrality measure for diverse types of networks that can accommodate an unlimited number of repeated interactions between nodes (actors). Most traditional measures are more appropriately suited for the transmission of valuable goods such as information and assume only optimal and efficient spreading between rational actors; however, our proposed measure is appropriate for disease transmission and web surfing where repeated back-and-forth interaction is the rule rather than the exception. In addition, this measure is well suited for the transmission of information of which the cost is negligible and of which the trustworthiness increases as a result of feedback provided during the back-and-forth transmission process (e.g., the spread of microfinance, fashion, and SNS usage).

The diffusion process of diverse social objects has long been the subject of many network studies. Various social objects have been studied for their diffusion dynamics; examples include new medical innovations such as tetracycline [1, 2], attitudes or opinions [3], information [4, 5], participation in collective activities [6, 7], and infectious diseases [8–13]. Many studies have focused on the role of bridges or brokers that connect otherwise disconnected sub-populations and found that bridging populations are essential for the outbreak of infectious disease epidemics such as STDs. In the absence of bridging ties, many diseases would remain inside of a small core group, which would contain the infection because it would not be able to travel from the core to the general population [14–16]. Thus, it is crucial to identify this bridging population in the early stages of epidemics for network intervention.

**Data Availability Statement:** The data underlying the results presented in the study are available from https://osf.io/qu57w/.

**Funding:** This work was supported by the Social Science Korea (SSK) Project through the National Research Foundation of Korea funded by the

Ministry of Education, Republic of Korea (NRF-2017S1A3A2067165).

**Competing interests:** The authors have declared that no competing interests exist.

In a strict sense, the term bridge refers to a line, of which the removal leaves more components than when it is included in graph theory, i.e., it fragments the network. The corresponding concept for a node is a cutpoint. A cutpoint is a node of which the deletion dissects the remaining graph into two or more disconnected pieces [17]. However, the term bridge is closer to everyday usage and has already been used to refer to a set of people (rather than a line or lines) in many studies on STDs [16, 18–20]. In this paper, we use the term bridging activity, or the role of the bridge, to refer to the structural potential to connect two nodes in a network.

Although existing studies have proposed a variety of ways to measure the extent to which each actor plays the role of a bridge in a network, most assume that diffusion processes take place on the basis of network paths that do not allow repeated exposure. In this paper, we propose that considering walks, rather than paths, provides a more effective network measure to identify nodes with high bridging potential. Our approach entailed the application of our proposed measures to two types of diffusion processes: the transmission of syphilis in Chicago area and the diffusion of microfinance in 43 rural Indian villages. We first discuss a few of the most widely used measures to explain the difference between our proposed new measure and previous measures. The purpose of this discussion is not to extensively examine these previous measures; instead, we limit our discussion to highlight the differences between the newly proposed measure and previous measures.

## Existing approaches to measure bridging positions

To highlight the uniqueness of our proposed measure, we first introduce the following three different concepts of a connected structure in a network: walk, trail, and path. Assume that a network consists of points (or actors) and lines that link two different points. A *walk* is an alternating sequence of points and lines beginning and ending with points in which each line is incident with the two points immediately preceding and following it. A walk can include identical points or lines multiple times; the dispersion (or flow) can go back and forth through the same points or lines without any limitation. A walk is termed a *trail* if all the lines are distinct and a *path* if all the points, and necessarily all the lines, are distinct (Harary 1969: 13) [21]. The following graph illustrates this difference.

In Fig 1, *ABEBC* is a walk that is neither a trail nor a path because two of the lines are not distinct, i.e., line B to E and line E to B. In contrast, *BEGFDE* is a walk and a trail but not a path because point E appears twice. *BEGH* is a walk and a trail but also qualifies as a path because no lines or points are traversed multiple times. As we discuss below, most traditional measures are based on paths; however, our new measure, namely walk-betweenness, is based on entire walks in the network.

Perhaps the most widely used measure of a bridging position is the betweenness centrality introduced by the seminal work of Freeman (1978) [22]. As Freeman summarized, the betweenness centrality of node *i* is the number of shortest paths that pass through node *i* linking other two nodes in the network. When a person is strategically located along the paths linking two other people, that person is in a potential position to control the communication transmission or coordinate group activities, and thus, exert influence or gain power (1978: 221) [22].

If the object of diffusion is a valuable resource and if the agents of the diffusion do not want to waste time or energy to repeat nodes or lines, it would be natural to assume paths rather than trails or walks. Furthermore, shortest paths can be used to examine diffusion processes if the actors have adequate knowledge of the structure of the global network and are able to choose the shortest path. For example, when actors are seeking a piece of imminently necessary information, they prefer to traverse only the shortest paths; thus, people who are occupying bridging positions with high geodesic-betweenness scores obtain influential power.

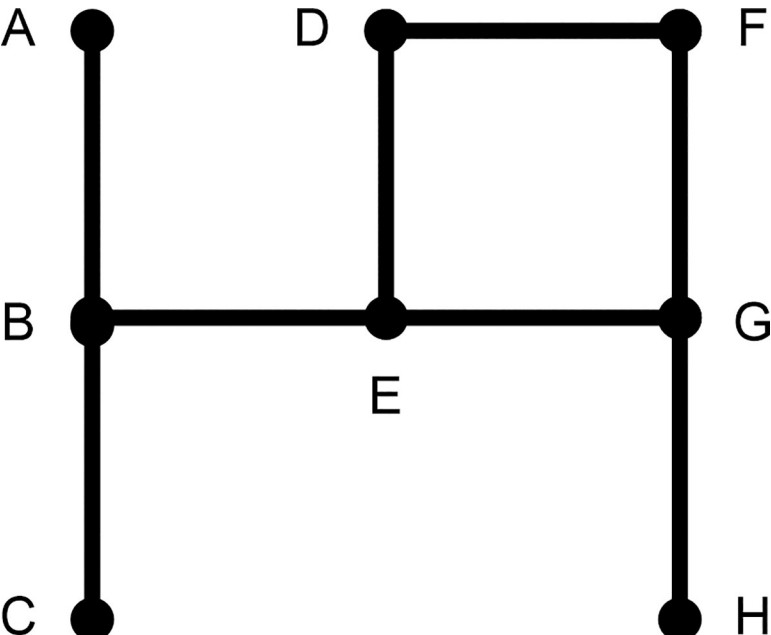

**Fig 1. Hypothetical network illustrating the differences between walks, trails, and paths.**

The underlying assumptions behind path-based network measures include that actors (1) desire the fastest flow (intention), (2) have adequate information about which path is the shortest (knowledge), or (3) are able to choose the shortest path (capability). However, these assumptions are too strict. Several approaches have been proposed to address these limitations. Freeman and his colleagues proposed another betweenness measure known as flow betweenness based on Ford and Fulkerson's (1956) concept of network flows [23, 24]. Betweenness centrality calculates the number of times a node is located on the shortest path between the source (or transmitter) nodes and target (or sink) nodes, whereas flow betweenness measures the proportion of the number of flows that are required to pass through the node to achieve the maximum number of flows between the sources and targets. The former is essential when it is desirable for messages (or products) to be diffused as fast as possible; the latter becomes crucial when the goal is for messages to spread out as much as possible. Achieving the maximum flow sometimes requires the flow to proceed along non-geodesic paths. However, flow betweenness still assumes the existence of only paths instead of trails or walks.

Another popular approach is to focus on a bridging actor who links two other actors that would not otherwise be connected. In this regard, the brokerage measure tends to emphasize political influence from the brokerage role [25, 26], whereas the structural hole measure tends to evaluate the increase in information access and economic performance from non-redundant ties [27]. Unlike the previously mentioned betweenness measures that consider the entire (global) network when measuring the bridging position of each party, these approaches limit their attention to local networks. The brokerage measure only takes direct ties into account for measuring brokerage, whereas the structural hole measure, which is based on the number of non-redundant ties, considers direct ties and indirect ties with a length of two.

Similar to the two betweenness centralities examined initially, trails and walks are not taken into account for this measurement. This is reasonable since securing power or acquiring information is usually based on the recognition of people close to the actors. In other words, an actor's power (or ability to attain crucial information) is systematically affected by his or her

(direct or indirect) local ties' appreciation of the actor's bridging activities. For example, a person whose distance from the actor is ten usually does not play a role in obtaining crucial information in time. In contrast, betweenness centralities measure a bridging position from a global point-of-view; they measure the bridging position of each actor when the entire network is taken into account rather than simply the local networks around the actor.

## When do we need a walk-based approach?

Now, assume that transmission costs are negligible or that the value does not arise from the monopoly of the transmitted object; in this case, transmission is not based solely on rationality, and thus, diffusions do not occur only on paths. Consider the spread of information regarding the microfinance program in developing countries for example. It may seem advantageous to use trails instead of paths to model the diffusion of microfinance because the same information may reach a given actor more than once from a different source; however, the source is unlikely to hear the information from the same person repeatedly. This is why Borgatti called for a new measure of centrality that is not based on paths [28]. Furthermore, the development of building trust with and persuading potential participants for microfinance may require repeated back-and-forth communications and thus, might happen on walks rather than trails. Another example in which a walk is appropriate is disease transmission; in particular, a person can be repeatedly re-infected by bacterial STDs from the same sexual partner. Furthermore, if our unit of analysis is a group or community instead of individuals, we can assume that most types of transmission will occur repeatedly between groups or communities. In addition to STDs, other examples include trade, phone traffic, and web surfing, which may take place between the same cities repeatedly in a given time frame.

This motivated us to propose a new measure of bridging activity based on walks rather than trails or paths. Centrality measures based on walks have already been proposed; however, they are not specifically designed to measure bridging positions. Perhaps the most widely used centrality measure is Bonacich's power centrality. Unlike betweenness centrality or non-redundant ties, Bonacich's power score takes all walks into account. Bonacich first proposed the measure known as eigenvector centrality, where an actor's centrality is their summed connections to others weighted by their centralities [29]. Later, he generalized this measure by adding the parameter $\beta$ that specifies the direction and extent of influence of other people's centralities that are directly or indirectly connected to the actor [30].

When we try to apply the Bonacich's power centrality to measure bridging activities, a couple of limitations appear. The value of $\beta$ should be assigned by researchers and the best way to choose and interpret a specific value of β remains open to debate [31–33]. This is unavoidable in the sense that the general nature of this measure requires that, depending on the network properties and analysis goals, $\beta$ must be able to change its sign and magnitude. Also, even though the magnitude and sign of $\beta$ are meaningful, the specific value of $\beta$ has no straightforward interpretation for bridging activities; for example, we do not have an intuitive understanding when the value of $\beta$ is 0.5.

Other centrality measures have been proposed that consider walks rather than paths or trails, such as the measures proposed by Friedkin (1998), Noh and Rieger (2004), Stephenson and Zelen (1989), and Yamaguchi (1994) [3, 34–36]. Again, these measures are not specifically designed to measure bridging activities. The most notable exception is Newman's proposal of betweenness centrality based on random walks. His betweenness was defined as the net number of times a walk passes through an actor [37]. Based on the analogy with electric current flow, he made the case for walks (rather than paths or trails) to measure bridging activities and proposed that the betweenness of node $i$ is equal to the number of times the diffusion passes

through $i$ on its random (a line chosen at random from the given possibilities) walks, averaged over all possible pairs of source nodes and target nodes. There is one important proviso though; this measure counts only the net number of times rather than the total number of times. He elaborated on the net number of times stating that [37]

> it would be perfectly possible for a vertex to accrue a high betweenness score if a random walk were simply to walk back and forth through that vertex many times, without actually going anywhere. . . . By "net" we mean that if a walk passes through a vertex and then later passes back through it in the opposite direction, the two cancel out, and there is no contribution to the betweenness. Furthermore, if, when averaged over many possible realizations of the walk in question, we find that the walk is equally likely to pass in either direction through a vertex, then again the two directions cancel.

We generally agree with Newman's proposal, especially the need for measures based on walks rather than paths or trails for certain types of diffusions. We, however, disagree with the idea that we should cancel out back-and-forth transmission when measuring the bridging potential. Unlike electric current flow where diffusion occurs almost instantly and the current in the network as an entirety is always conserved, if diffusion weakens or strengthens throughout the networks over time, the total count not net count may be meaningful.

Namely, if the object of diffusion is a transmittable disease (or rumor), back-and-forth transmission in opposite directions through a node may help the disease (or rumor) to survive and to remain in the population for a given longer time period, thereby enabling transmission from the source to the target. For this type of diffusion, we believe that, even if the diffusion were to pass through the node one way and return to the same node via another route sometime later, the node would still be engaging in bridging activity. This becomes more evident if we assume the unit of analysis to be a group, community, or country rather than an individual. For example, two communities in a certain area can transmit the Ebola virus to each other through back-and-forth transmission, thus perpetuating the survival of the virus for an extended period of time. As a result, the virus could be transmitted to a wider population outside the two communities.

Therefore, we propose a measure of bridging activities—based on walks rather than paths or trails—including back-and-forth transmissions through a node from opposite directions. Canceling happens only when a certain node transmits the information directly to itself because self-selection usually is not believed to contribute to the bridging activity, and thus, must not be counted. However, certain studies may still want to include self-selection in the calculation, especially if the unit of analysis is a group of people instead of individuals. In that case, self-selection does not mean returning to identical individuals. The R programming code for our measure, which includes self-selection, is available upon request. This measure is based on a simple and intuitive concept of bridging activity; the bridging activity of node $k$ in the diffusion from starting node $i$ to the target node $j$ is the complementary probability of the event that $i$ can reach $j$ without visiting $k$. The amount of bridging activities of $k$ that contribute to the flow from $i$ to $j$ is given by $(1-_kf^*_{ij})$, where $_kf^*_{ij}$ is the probability that $i$ can reach $j$ without visiting $k$ in a Markov process that permits every possible walk except for self-selection. We refer to this measure as walk-betweenness.

This measure was briefly introduced in a previous study [38] without any elaborated mathematical explanation. The paper advances the discussion of walk-betweeness measure in three major ways. First, it provides the full mathematical equations behind the measure and lay out detailed calculations readers need to use to measure the walk-betweeness for themselves. Second, it introduces a new usage of walk-betweeness: a cluster analysis based on transmission coefficients. This cluster analysis can identify non-adjacent or distant neighborhoods who are

close to each other in transmission paths. This could be essential to come up with a cooperative preventive strategy of local health-related organizations for transmittable diseases. Finally, it presents empirical tests of walk-betweeness measure by utilizing two new data sets: Syphilis transmission in Chicago area and microfinance diffusion in rural Indian villages.

## Materials and methods

### Proposal of walk-betweenness

We return to the first example of a hypothetical network to introduce our new measure. Fig 2 adds diffusion probabilities to Fig 1. For illustration purposes, we assume that diffusion takes place along ties with identical probabilities. Actor B (or community B) has three actors (or communities) to transmit to; thus, a one-third probability exists for each (actors A, C, and E). These probabilities can be changed as long as their sum is unity including self-selection; actors must have the potential to transmit the object to somebody else including themselves. We can express these probabilities as a matrix known as the transition matrix: $T = [t_{ij}]$, where $t_{ij}$ is the probability that an object diffuses from $i$ to $j$. All entries in the transition matrix must be non-negative, and the sum of the entries in each row must be unity. This transition matrix is associated with random walk and also known as a matrix of a Markov chain [34].

Let $\lambda_0 = [p_1, p_2, \ldots, p_n]$ (where $\sum_{i=1}^{n} p_i = 1$) be a row vector, each element of which represents the initial proportional distribution of objects such as diseases or information across the actors (or communities). Then $\lambda_0 T = \lambda_1$ becomes the new proportion across the actors after one period of transmission. The sequence of probability states can be induced in the same manner: $\lambda_1 = \lambda_0 T$, $\lambda_2 = \lambda_1 T = (\lambda_0 T)T$, $\lambda_3 = \lambda_2 T = (\lambda_0 T^2)T$, and $\lambda_n = \lambda_{n-1} T = (\lambda_0 T^{n-1}) = \lambda_0 T^n$. A Markov chain that has a $T^n$ term with no zero entries for some integer $n$ is known as a regular Markov chain. We can show that the powers of $T^n$ approach a probability transition matrix $L$,

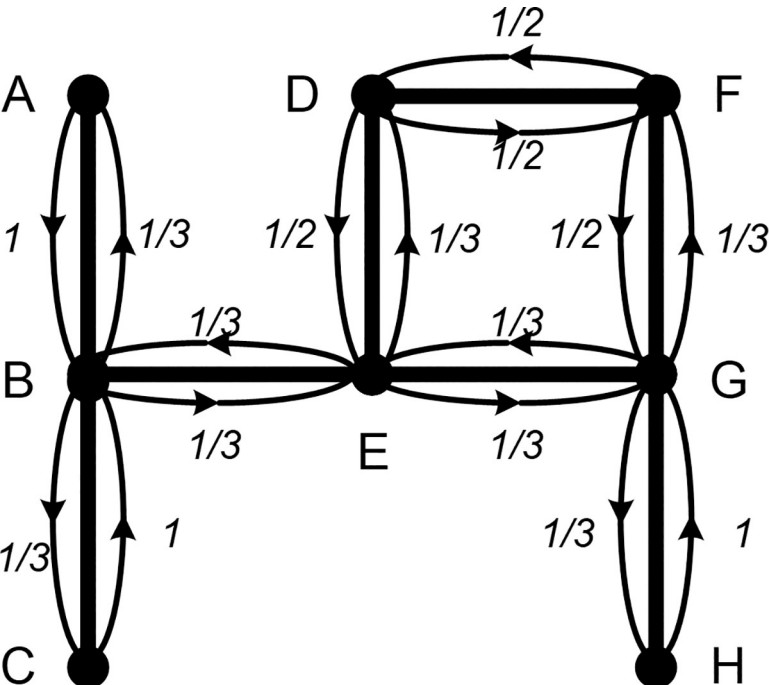

**Fig 2. Hypothetical network indicating the transition probabilities.**

where each row of $L$ is the same probability row vector $\alpha$, and that its components are all positive (Kemeny and Snell 1960; 70) [39]. This ensures that $\lambda_n$ converges to $\lambda_0 T^n = \lambda_0 L = \alpha$ as following, where $\alpha$ is a limiting probability that shows a steady-state proportion.

$$\lambda_0 T^n = \lambda_0 L = [p_1, p_2, ..., p_n] \begin{bmatrix} \alpha_1, \alpha_2, & \cdots & , \alpha_n \\ \vdots & \ddots & \vdots \\ \alpha_1, \alpha_2, & \cdots & , \alpha_n \end{bmatrix} = [\sum_{i=1}^{n} p_i \alpha_1, \sum_{i=1}^{n} p_i \alpha_2, ..., \sum_{i=1}^{n} p_i \alpha_n]$$

$$= [\alpha_1, \alpha_2, ..., \alpha_n] (\because \sum_{i=1}^{n} p_i = 1) = \alpha$$

Thus, if a Markov chain is regular, i.e., some power of its transition matrix has only positive entries, then, there should exist a stationary distribution of the probability state among actors, regardless of the initial distribution, $p_i$.

**Measuring walk-betweenness.** For our purpose of measuring bridging activities, one of the most interesting quantities in a Markov chain is the mean first passage time. Let $f_{ij}^n$ be the probability that the Markov chain will be in state $j$ for the first time at exactly the $n$-th step, given that it starts from $i$; this is the probability of the first passage time. This measure leads us to two other important quantities: $f_{ij}^*$ and $m_{ij}$. If $f_{ij}^n$ is summed up to infinite steps, $\sum_{n=1}^{\infty} f_{ij}^{(n)} = f_{ij}^*$ is the probability that the Markov chain will be in state $j$ at least once given that it starts from state $i$. Moreover, this implies that the mean first passage time from $i$ to $j$ is $m_{ij} = \sum_{n=1}^{\infty} n f_{ij}^{(n)}$. It is also possible to measure the probability that the diffusion from node (actor) $i$ reaches node (actor) $j$ without visiting node (actor) $k$, i.e., $_k f_{ij}^*$ (Chung 1967: especially Chapter 11) [40]. Here, $_k f_{ij}^*$ ranges from zero to one and has an intuitive interpretation. If $_k f_{ij}^*$ is zero, then we conclude that it is impossible for the diffusion to pass from $i$ to $j$ without passing through $k$, i.e., $k$ is essential in the transmission process from $i$ to $j$. In contrast, if $_k f_{ij}^*$ equals one, $k$ is of no importance in the diffusion development from $i$ to $j$. Thus, the complementary probability ($1 - _k f_{ij}^*$) is the measure that quantifies the bridging activity of actor $k$ in the diffusion from $i$ to $j$; its value is maximal ($1 - _k f_{ij}^* = 1$) when the diffusion is not possible from $i$ to $j$ without actor $k$, and it is minimal ($1 - _k f_{ij}^* = 0$) when it is always possible to transmit information from $i$ to $j$ without passing through $k$. We believe this is an intuitive concept of bridging activity. This probability can be summed for every possible directed tie from $i$ to $j$ to measure the walk-betweenness of actor $k$ as follows:

$$WB_k = \sum_{i=1}^{n} \sum_{j=1}^{n} (1 - _k f_{ij}^*)(i \neq j \neq k).$$

Here, $_k f_{ij}^* = \frac{m_{ik} + m_{kj} - m_{ij}}{m_{jk} + m_{kj}}$ (Chung 1967; 65) [40]; the mean first passage time matrix $\mathbf{M}$ is given by $\mathbf{M} = (\mathbf{I} - \mathbf{Z} + \mathbf{E}\mathbf{Z}_{dg})\mathbf{D}$, where $\mathbf{I}$ is the identity matrix, $\mathbf{Z} = (\mathbf{I} - (\mathbf{T} - \mathbf{A}))^{-1}$, $\mathbf{T}$ is a transition matrix, $\mathbf{A}$ is a matrix where each column is $\alpha$, $\mathbf{E}$ is an $n \times n$ matrix of which all the elements are 1, $\mathbf{Z}_{dg}$ is the diagonal matrix of $\mathbf{Z}$, and $\mathbf{D}$ is the diagonal matrix of which the diagonal elements are given by $d_{ii} = 1/\alpha_i$ (Kemeny and Snell 1960; 79) [39]. $WB_k$ measures the average probability that a source passes through node (actor) $k$ in the diffusion process to a target in the network. For example, if the value of the expression is 0.5, on average, half of the diffusion passes

**Table 1. Comparison of the measures of bridging positions of nodes in a hypothetical network in Fig 1.**

|   | Betweenness centrality[a] | Flow Betweenness | Non-redundant ties | Bonacich Index[b] | Newman Betweenness | Walk Betweenness |
|---|---|---|---|---|---|---|
| A | 0 | 0 | 1 | 0.05 | 0.25 | 0.31 |
| B | 11 | 11 | 3 | 0.12 | 0.64 | 0.65 |
| C | 0 | 0 | 1 | 0.05 | 0.25 | 0.31 |
| D | 2 | 3 | 2 | 0.16 | 0.41 | 0.53 |
| E | 13 | 15 | 3 | 0.20 | 0.74 | 0.77 |
| F | 1 | 3 | 2 | 0.15 | 0.39 | 0.50 |
| G | 8 | 9 | 3 | 0.19 | 0.58 | 0.64 |
| H | 0 | 0 | 1 | 0.08 | 0.25 | 0.30 |

Note

a: The network data were binarized to calculate the betweenness score.

b: $\beta$ is set to 0.5 for the calculation.

through node $k$ in the network. In addition to providing the walk-betweenness measure, the mean first passage time presents another useful piece of information; specifically, the correlation between nodes in diffusion processes. We would often like to know which actors (or groups) are close to one another in the course of transmission. For example, it would be useful to know that the more often a process passes through actor A, the less often (or more often) it passes through actor B. The following equation calculates the correlation among actors (or groups) and ranges from −1 to 1 (Kemeny and Snell 1960; 84–5) [39]. In this paper, we refer to this as the transmission correlation

$$C_{ij} = \alpha_i z_{ij} + \alpha_j z_{ji} - \alpha_i d_{ij} - \alpha_j \alpha_i.$$

These coefficients can be interpreted in a way similar to the traditional Fisher's correlation coefficients in statistics. Based on the above correlation matrix, we can divide the actors (communities) into several groups based on their walk proximity to each other along the route of transmission. Cluster analysis is an appropriate tool for the purpose of creating sub-groups based on the correlation matrix. Later, we demonstrate the use of this transmission correlation matrix for clustering Chicago-area communities in terms of the diffusion of syphilis and discuss the possible policy implications for preventive strategies.

**Characteristics of walk-betweenness.** Table 1 summarizes the scores of different measures for the network in Fig 2 and allows the characteristics of walk-betweenness to be examined. As Newman pointed out (2005, 10), various centrality measures are known to be strongly correlated to each other, and we want to focus on specific nodes for which different measures produce very different scores to evaluate the utility of the measures [37]. The most common feature all measures share is that actors, B, E, and G are the most central actors, although the orders between them are different for each measure. Upon closer examination, however, the differences between measures are revealed. The betweenness centrality score for actor B is higher than that for actor G because actor B belongs to the only path between actors A and C, whereas actor G is along the only path for actor H; there exists an alternative path for F through D in addition to the path through G. Flow betweenness produces similar scores. However, because flow betweenness focuses on maximum flow rather than fastest flow, the score for actor F is three times higher, specifically, it increases from 1 to 3. To achieve maximum flow, we would need to pass through actor F, who is in a peripheral location, more frequently. Unlike betweenness centrality or flow betweenness, the number of non-redundant ties is identical for actors B, E, and G because when we consider only direct and indirect ties with a length

of two, there is no redundancy between the direct ties of these three actors. When $\beta$ is set to 0.5 for the calculation of Bonacich's index, actor E's score is the highest at 0.2.

The scores of Newman's betweenness or walk-betweenness are also highest for actors B, E, and G. Other actors, however, especially those who occupy the most marginal positions such as actors A, C, F, and H, now have relatively high scores compared to other previous measures. For example, actor A is never on the shortest paths between other actors; thus, its betweenness centrality score is 0 whereas its score increases to 0.25 and 0.31 for Newman's betweenness and walk-betweenness, respectively, which is almost half of the score of actor G. The reason becomes clear when we consider the difference between paths and walks. When we calculate the betweenness centrality based on paths only, we only count the number of occurrences for which the actor exists on the shortest paths between other actors. In contrast, in principle, all possible walks are counted for Newman's betweenness or walk-betweenness; thus, actors who occupy peripheral positions such as actors A, C, F, and H become more central. If the diffusion dynamics are not based on efficiency-seeking dynamics using scarce resources, Newman's betweenness or walk-betweenness is preferable to betweenness centrality.

More specifically, the diffusion dynamics more closely approximate walks rather than paths if (1) actors' diffusion behaviors are not based on the rationality of efficient transmission (for example, transmitting infectious diseases through sexual contacts or engaging in web surfing for fun with little cost), (2) actors do not want to follow the shortest paths despite having full knowledge of the network structure or the capability to follow the shortest path, (3) the unit of analysis is a group, instead of individuals, such as communities where the transmission process does not follow the efficiency principle, or (4) the focus of the analysis is not on future diffusion dynamics but on past events. For example, if the probabilities given in Fig 2 correspond to actual data collected over the past six months, we can conclude that one third of the diffusion that passes through actor B went to actor A. The greater the extent of inefficiency of the transmission process with complete information, the more important marginal actors such as A become.

We now examine the difference between Newman's betweenness and walk-betweenness. The fact that the scores of walk-betweenness are larger than those of Newman's betweenness is to be expected. This is because walk-betweenness also counts back-and-forth transmissions through a node that are disregarded in Newman's measure, which considers only the net amount; thus, in general, it is larger than Newman's betweenness. The largest increases are observed for actors D and F; their scores rise by approximately 30%. In Fig 2, actors D and F provide many back-and-forth transmissions through actors E and G, and thus, their scores of walk-betweenness are greater compared to Newman's scores.

As previously mentioned, the mean first passage time also provides correlation coefficients among actors. Table 2 lists the correlation coefficients among the eight actors shown in Fig 2.

**Table 2. The correlation matrix between nodes in a hypothetical network in Fig 1 provided by our walk betweenness approach.**

|   | A | B | C | D | E | F | G | H |
|---|---|---|---|---|---|---|---|---|
| A | 1.00 | 0.84 | 0.51 | −0.54 | −0.29 | −0.64 | −0.69 | −0.48 |
| B | 0.84 | 1.00 | 0.84 | −0.63 | −0.15 | −0.79 | −0.83 | −0.59 |
| C | 0.51 | 0.84 | 1.00 | −0.54 | −0.29 | −0.64 | −0.69 | −0.48 |
| D | −0.54 | −0.63 | −0.54 | 1.00 | 0.30 | 0.67 | 0.09 | −0.07 |
| E | −0.29 | −0.15 | −0.29 | 0.30 | 1.00 | −0.17 | −0.02 | −0.22 |
| F | −0.64 | −0.79 | −0.64 | 0.67 | −0.17 | 1.00 | 0.53 | 0.21 |
| G | −0.69 | −0.83 | −0.69 | 0.09 | −0.02 | 0.53 | 1.00 | 0.80 |
| H | −0.48 | −0.59 | −0.48 | −0.07 | −0.22 | 0.21 | 0.80 | 1.00 |

The coefficient value of 0.84 between actor A and actor B indicates that the more often the transmission process passes through actor A, the more often it will also pass through actor B. Three actors (A, B, and C) have positive coefficients between them but have negative coefficients between other actors. This implies that the three actors (A, B, and C) can be treated as one distinct group from the others in the transmission process. For example, in terms of infectious diseases, once area A is struck by a disease, areas B and C become highly vulnerable. Cluster analysis performs this type of partitioning systematically. Fig 3 shows one result of cluster analysis of the network in Fig 2 using UCINET 6 [41].

The cluster is formed on the basis of single-link criteria by which the distance between two clusters is defined as the largest similarity (in our case, correlation) between actors [42]. As the figure reveals, actors A, B, and C can instantly be defined as one group at the level of the 0.84 coefficient or the high walk proximity to each other (the largest coefficient within the three actors is 0.84), and actors G and H form another distinct cluster at the level of the 0.80 coefficient, followed by actors D and F. If we relax the criteria to 0.30, two groups emerge in the course of diffusion: actors A, B, and C form one group and the other actors form another group. To illustrate and discuss the usability of walk-betweenness, we apply it to the diffusion of syphilis in the Chicago area and the diffusion of microfinance in rural Indian villages. Note that these two empirical analyses must be treated as illustrations for revealing the usefulness of our walk-betweeness for specific empirical cases.

## Study 1. Diffusion of syphilis in the Chicago area

STDs are a major public health challenge in the United States. The Center for Disease Control and Prevention (CDC) estimates that 20 million new infections occur each year [43]. In addition to their physical and psychological consequences, STDs are also a serious drain on the United States health care system, costing the nation 15.6 billion dollars in total lifetime direct medical expenses in 2008 [44]. Furthermore, various STDs may lead to cancer, infertility, ectopic pregnancies, spontaneous abortion, and stillbirth, in addition to low birth weight for infants and an increased risk for contracting the Human Immunodeficiency Virus (HIV) [45].

```
 Level     A B C E D F G H
───────   _ _ _ _ _ _ _ _
 0.84     XXXXX . . . . .
 0.80     XXXXX . . . XXX
 0.67     XXXXX . XXX XXX
 0.53     XXXXX . XXXXXXX
 0.30     XXXXX XXXXXXXXX
−0.15     XXXXXXXXXXXXXXX
```

Fig 3. **Results of a hierarchical cluster analysis of the hypothetical network.**

Syphilis, a genital ulcerative disease, is highly infectious but easily curable in its early (primary and secondary) stages. If untreated, it can lead to serious long-term complications, including neurologic, cardiovascular, and organ damage, and even death. Congenital syphilis can cause stillbirth, death soon after birth, and physical deformities and neurological complications in children that survive. Syphilis, as for many other STDs, facilitates the spread of HIV, increasing transmission of the virus at least two- to five-fold [46]. The rate of primary and secondary (P&S) syphilis reported in the United States decreased during the 1990s; in 2000, the rate was the lowest since reporting began in 1941. Between 2005 and 2013, the national P&S syphilis rate nearly doubled from 2.9 to 5.3 cases per 100,000 people, the highest since 1995, i.e., the number of reported P&S syphilis cases in the United States increased from 8,724 to 16,663 [47, 48].

**Bridging positions in sexually transmitted disease studies.** Historically, the main focus of STD epidemiology has been on the attributes and behaviors of individuals, which is consistent with the dominant perspectives in clinical medicine, chronic disease epidemiology, and psychology [8]. However, since the mid-1980s, many researchers have reconsidered the important role of sexual networks in sustaining the extraordinarily high infection rates in the United States. Unlike chronic diseases, the odds of being infected and infecting others is determined by factors beyond the level of the individual; that is, they depend not only on the individual person's risk factors but also on the risk factors of that person's sexual partners [49]. Thus, applying social network analysis to STD transmission dynamics has steadily gained research attention [11, 38, 50–53]. However, this type of analysis has two limitations. First, the level of analysis in most studies is limited to dyadic. Based on the contact tracing method [50] or studies of residential areas [54, 55], most studies have focused on the dyadic relationship between infected people and their partners. They rarely aggregated the data to the level of communities, and thus, could not identify the risk factors that exist at the group-level beyond individual attributes or the dyadic relationship. Second, few studies have focused on the existence of bridges between distinct sub-populations. Even though community A may have a higher prevalence rate than community B, community B can be a much more efficient (powerful) transmitter of infection if it maintains sexual partners in distinct sub-populations, thereby providing a distinctive link or bridge for infection to spread between the two sub-populations rarely connected otherwise. In order to identify this type of bridges of STD transmission, it becomes evident that the walk betweeness that allows any number of repeated transmissions between residential areas is more conceptually proper than other traditional centrality measures.

**Data.** STD specialty clinics use the contact-tracing method to trace sexual partners of infected people. They maintain two types of records during tracing. First, once a person is diagnosed at a clinic or referred from a hospital or doctor, a disease intervention specialist (DIS) conducts an interview, which is recorded in an interview record (IR). During the interview, the infected person is asked to enumerate their sexual partners with locating information to enable the DIS to contact them. A field record (FR), which contains the locating information, is completed for each sexual partner of the infected individual. The DIS uses the FR to contact the sexual partner to recommend STD testing. A sexual partner who is determined to have been infected, is interviewed in depth using an IR, and during the interview, location information about their sexual partners is obtained. This tracing continues until no infected partners are found. The city of Chicago operates freestanding STD specialty clinics that provide full-service STD diagnostic and treatment services free of charge. These clinics are staffed by clerks, approachable DISs, laboratory technicians, nursing practitioners, and physicians. The clinics are located in high-risk areas of the city.

By combining these two types of records (IR and FR) from all eight STD specialty clinics in Chicago, geo-coding at the census tract level for each partnership that were infected by syphilis in 2000 and 2001 was obtained. The data was collected based on the support from the School of Public Health of the University of Illinois at Chicago in 2002. The raw data was provided at the aggregate level only without any individual identifier.

Although these data are useful in the sense that they provide a sufficient number of infected people compared to a survey of the general population that only yields a handful of infected people (according to the Chicago Health and Social Life Survey, only 2% of the general population were infected by any STDs in the past 12 months, Youm and Laumann 2002) [53], they have limitations. To ensure privacy, people of higher socioeconomic status (SES) often consult private doctors who are likely to place a lower priority on reporting STDs to government agencies. Thus, the conclusions of our study are slightly biased toward low SES individuals that are more likely to visit clinics for free diagnosis and treatments.

**Network data construction.** The Chicago area consists of 77 communities defined by census tracts. Each cell in an initial $77 \times 77$ matrix represents the number of partnerships between two communities. For example, we recorded 5 in the (3,6) cell in the matrix if a person who lived in community 3 reported five partners who resided in community 6 in 2001 and 2002. The diagonal cells indicated the number of partnerships within the community. Because sexual ties are necessarily mutual (if person A in community 1 has two sexual partners in community 2, there must be two people in community 2 who have sexual partnerships with person A), we summed the initial matrix and its transposed matrix to make it symmetric. Now, the final matrix represents the number of sexual matching ties between the 77 communities, where at least one of the partners was infected by syphilis, and contains a total of 568 sexual relationships.

**Bridging communities with low infection rates: Hidden bridging communities.** Many bridging communities in STD dynamics have been largely ignored by traditional preventive efforts because of their own low infection rates, although they may require special preventive efforts because of their essential role in transmitting STDs to the entire population. Fig 4 provides a combined view of bridging activities and infection rates for 56 communities after we excluded 21 isolated communities that had no sexual ties to other communities from our analysis. The dotted lines in the figure denote the mean values. For example, the communities in the region on the upper right have higher-than-average incidence rates and higher-than-average walk-betweenness. We present plots using our walk betweenness score (Panel A) and Newman's betweenness score (Panel B).

We discover in Fig 4-Panel A the existence of hidden bridging communities, namely, bridging communities with low infection rates. Communities in the region on the upper left of Fig 4-Panel A, including communities 22, 23, and 48, exhibit above-average bridging behavior. However, less attention is paid to these communities due to their lower infection rates compared to communities 36 and 53 (communities in the region on the lower right of Fig 4, Panel A) that have high infection rates. Special attention must be given to these hidden bridging communities because they are crucial in transmitting STDs to the remaining population. We identified fourteen hidden communities (communities 22, 23, 24, 25, 26, 28, 33, 45, 46, 48, 49, 67, 69, and 73); these communities account for 25% of the total number of communities in Fig 4-Panel A. That said, Panel B shows that Newman's betweenness measure reveals fewer hidden bridging communities. This disparity comes from the difference in assumptions between two measures: Newman's betweenness cancel out back-and-forth opposite transmission through a certain community to calculate net count while walk-betweeness count all the transmissions to obtain total count. We believe that this difference brings a bigger variance to walk-betweeness

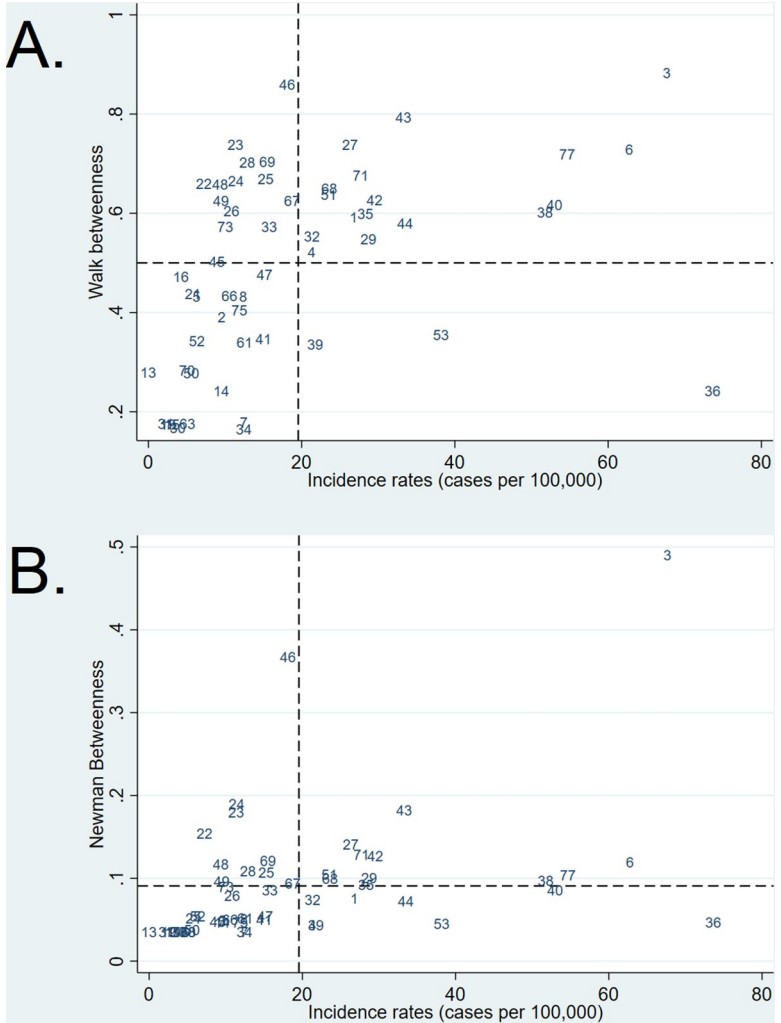

**Fig 4. Hidden bridging communities in Chicago.** Using the aggregated contract tracing data from STD specialty clinics in Chicago, we plot incidence rates of syphilis during the period 2001–2002 (on the x-axis) against scores of walk-betweenness and Newman betweenness (on the y-axis) to identify the hidden bridging communities (upper-left areas) that are priori given less attention.

compared to Neman's betweeness in general as shown in Fig 4, which could bring more hidden bridges.

**Cluster analysis of communities.** Another step toward an effective preventive strategy is to partition the entire population into smaller clusters of communities that are in close proximity with respect to transmission paths, and thus, more likely to transmit STDs to each other. Fig 5 was obtained by applying Johnson's clustering algorithm and defining the distance between clusters as the average similarity between communities based on the transmission correlation previously discussed [42]. In the hierarchical cluster analysis, we stopped merging communities when all of the communities were in the top 10% for bridging based on walk-betweenness. We tried several different cut-off points but the main conclusion, the existence of hidden bridging communities, remains.

Fig 5 shows a plot of the cluster membership on a map of the Chicago area, which reveals two important findings. First, in general, clusters are separated by racial/ethnic dimensions. For example, cluster 7 mainly consists of white communities whereas most communities in

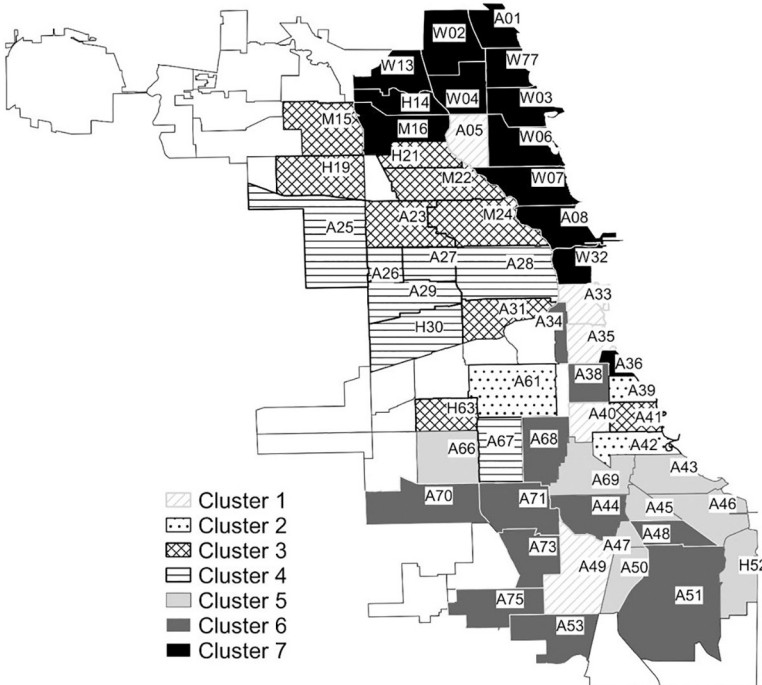

**Fig 5. Seven clusters of Chicago communities based on the incidence of syphilis transmission.** We color Chicago areas according to seven cluster solutions provided by our walk-betweenness approach. For each area, the first letter represents the racial/ethnic group that forms the majority in the community: A (African American), H (Hispanic), M (Mixed) (no single racial/ethnic group forms the majority), and W (white) and the next two-digit number represents the community number in the Chicago area. The map is from the Chicago Data Portal (https://data.cityofchicago.org/ Facilities-Geographic-Boundaries/Boundaries-Community-Areas-current-/cauq-8yn6). This site provides applications using data that has been modified for use from its original source, www.cityofchicago.org, the official website of the City of Chicago. The City of Chicago makes no claims as to the content, accuracy, timeliness, or completeness of any of the data provided at this site. The data provided at this site is subject to change at any time. It is understood that the data provided at this site is being used at one's own risk.

cluster 5 and 6 are African Americans. This clearly shows that sexual ties in the Chicago area were built based on racial/ethnic identities. Second, although most communities in a cluster are geographically close, certain exceptions can be found. For example, in cluster 1, although communities 5, 40, and 49 are to a greater or lesser extent distant from other communities in the same cluster, they share close transmission passages even though they are not adjacent to each other. This is especially noteworthy because traditional joint preventive efforts have been primarily been organized between neighboring communities. Cluster analysis based on walk-betweenness suggests the necessity of joint preventive efforts between geographically distant communities within the same cluster.

## Study 2. Diffusion of microfinance in rural India

Thus far, we show how a walk-based conception of diffusion processes can help us better understand the disease transmission dynamics at the aggregated level. Here, we turn our attention to a widely cited diffusion study to illustrate the usefulness of our approach at the individual level. To examine the idea that the network position of early adopters shapes the diffusion outcomes in a community, Banerjee et al. (2013) designed an innovative study that collects information about social networks and tracked the participants in the microfinance program Bharatha Swamukti Samsthe (BSS) among all households in 43 villages [56]. To examine the

usefulness of our walk betweenness measure, we revisit this study data which is publicly available to researchers (https://dataverse.harvard.edu/dataset.xhtml?persistentId=hdl:1902.1/21538, last accessed 2019 October 10).

In each of the 43 villages, the BSS program was first introduced to the town leaders, who were asked to help organize a meeting at which their followers could be provided with information about microfinance. Using the variations in the network position of early adopters (in this context, village leaders), they found that villages with higher "diffusion centrality" of town leaders had significantly higher participation rates. The proposed diffusion centrality is defined as a centrality that considers the time flow in diffusion processes. Although their research found that diffusion centrality outperforms other traditional centrality measures, we argue that the walk-betweenness measure can more effectively capture the actual diffusion process for the following reasons.

First, the diffusion of microfinance may not follow the most efficient pathways. Like gossip, residents may spread and receive the information about microfinance through multiple channels, for example, a person may receive the information twice from different neighbors. As a result, the diffusion may occur not only on paths but through repeated exposure on walks. Furthermore, because the value of a finance program is uncertain and is contingent upon the joint liability of a small lending group, multiple exposure is essential for the successful diffusion of microfinance participation [57, 58]. Third, the diffusion of microfinance resembles a walk since positive or negative feedback on microfinance may flow through the same point and via the same route repeatedly. Back-and-forth feedback may create trust in the novel and unfamiliar information about microfinance and increase the likelihood of participation in the program. To test these ideas, we compared the performance of walk-betweenness to explain the diffusion of microfinance with other traditional centrality measures including diffusion centrality.

**Data.** We revisited data collected by Banerjee et al. (2013), who used surveys to collect information about the basic demographics and social networks in seventy-five villages in rural southern Karnataka before the BSS program was introduced [56]. The survey included a module to collect social network data across twelve dimensions of social ties: those who visit the respondent's home, those whose homes the respondent visits, kin in the village, nonrelatives with whom the respondent socializes, those from whom the respondent receives medical advice, those from whom the respondent would borrow money, those to whom the respondent would lend money, those from whom the respondent would borrow material goods (kerosene, rice, etc.), those to whom the respondent would lend material goods, those from whom the respondent gets advice, those to whom the respondent gives advice, and those with whom the respondent goes to pray (at a temple, church, or mosque) [56].

**Network data construction.** Following Banerjee et al. (2013), we constructed household-level social networks for forty-three villages where BSS programs were actually in operation [56]. In each social network, the nodes are households and the edges indicate that at least one social tie of the aforementioned dimensions exists between two households in a binary network. Formally, a binary network is expressed as a matrix $B = [b_{ij}]$, where $b_{ij} = 1$ if at least one type of relationships exists between household $i$ and $j$ and $b_{ij} = 0$ otherwise. Based on $B$, a transition matrix T can be created: $T = [t_{ij}]$, where $t_{ij} = \frac{b_{ij}}{\sum_j b_{ij}}$. We also constructed a weighted network in which each edge is weighted by the number of different types of social relationships among the twelve types between two households. Formally, a weighted network is expressed as a matrix $W = [w_{ij}]$, where $w_{ij}$ is the number of different types of social relationships between household $i$ and $j$. Based on $W$, a transition matrix $T$ can be created: $T = [t_{ij}]$, where $t_{ij} = \frac{w_{ij}}{\sum_j w_{ij}}$.

**Walk-betweenness and diffusion of microfinance.**    Using these two types of social net-
work data, we calculated the walk-betweenness and other centrality measures including the
diffusion centrality proposed by Banerjee et al (2013) [56]. Table 3 presents the descriptive sta-
tistics of leader households' walk-betweenness and other characteristics, and Table 4 shows all
pair-wise correlations between walk-betweenness and other measures.

Among the leader household sample, we estimated linear probability models to predict
whether one of the followers participates in the microfinance program. For each leader house-
hold, the participation is coded 1 if at least one follower participated in the microfinance pro-
gram and 0 otherwise. In all regression models, we control for household socio-economic
characteristics that are known to influence microfinance take-up, including the number of

**Table 3. Descriptive statistics of leader households' walk-betweenness and other characteristics.**

| Variable | Mean or % | SD |
|---|---|---|
| *Outcomes* | | |
| Alter participation rate of microfinance program | 77.6% | |
| Alter participation rate (excluding leader households) | 72.2% | |
| *Network positions* | | |
| Walk betweenness | .570 | .151 |
| Diffusion centrality | 5.348 | 4.314 |
| Degree centrality | 13.021 | 9.228 |
| Closeness centrality | .002 | .001 |
| Betweenness centrality | 349.549 | 576.730 |
| Eigenvector centrality | .300 | .235 |
| Walk betweenness (weighted) | .571 | .159 |
| Diffusion centrality (weighted) | 3.650 | 3.812 |
| Degree centrality (weighted) | 25.204 | 20.546 |
| Closeness centrality (weighted) | .001 | .001 |
| Betweenness centrality (weighted) | 410.571 | 655.042 |
| Eigenvector centrality (weighted) | .153 | .197 |
| *Control variables* | | |
| Number of rooms | 2.710 | 1.614 |
| Number of beds | 1.151 | 1.486 |
| Number of rooms per capita | .565 | .357 |
| Number of beds per capita | .231 | .304 |
| Electricity | | |
| Household has private electricity | 71.7% | |
| Government | 22.9% | |
| No electricity | 5.4% | |
| Latrine | | |
| Household has its own latrine | 40.9% | |
| Common | .5% | |
| No latrine | 58.6% | |
| Own participation of microfinance program | 23.6% | |
| Observation | 1,129 | |

Note. All network measures are calculated for all leader households based on the multilayered networks that span
across 12 different types of relations across all leaders and followers in a village. We consider within-household
relations (i.e., self-loop) as well. We present both results network centrality measures based on both binary and
weighted networks.

**Table 4. Correlations across seven centrality measures among leader households in 75 Indiana villages.**

| Binary network measures | (1) | (2) | (3) | (4) | (5) |
|---|---|---|---|---|---|
| (1) Walk betweenness | | | | | |
| (2) Diffusion centrality | .762 | | | | |
| (3) Degree centrality | .854 | .811 | | | |
| (4) Closeness centrality | .357 | .362 | .302 | | |
| (5) Betweenness centrality | .592 | .649 | .778 | .044 | |
| (6) Eigenvector centrality | .781 | .845 | .847 | .451 | .576 |
| Weighted network measures | (1) | (2) | (3) | (4) | (5) |
| (1) Walk betweenness (weighted) | | | | | |
| (2) Diffusion centrality (weighted) | .686 | | | | |
| (3) Degree centrality (weighted) | .801 | .784 | | | |
| (4) Closeness centrality (weighted) | .295 | .291 | .230 | | |
| (5) Betweenness centrality (weighted) | .524 | .457 | .607 | .048 | |
| (6) Eigenvector centrality (weighted) | .586 | .858 | .697 | .322 | .355 |

Note. All network measures are calculated for all leader households based on the multilayered networks that span across 12 different types of relations across all leaders and followers in a village. We consider within-household relations (i.e., self-loop) as well. The binary network is identified if two village residents are connected through at least one type of relationship, and the weighted network is constructed by taking the sum of all overlapping relationships.

rooms, beds, rooms per capita, beds per capita, electricity (private/government/no electricity), and, latrine ownership (own/common/no latrine). We account for the leader's own participation in the microfinance program. Also, we included fixed effects associated with each village.

In Table 5, Model 1 shows that an one standard deviation increase of leaders' walk-betweenness score leads to a 17 percentage point increase in the probability of their alters' participation. In the same vein, Model 2 shows that "diffusion centrality" is positively associated with alters' participation, which is consistent with the results obtained by Banerjee et al. (2013) [56]. In Model 3, we find that degree centrality is also positively associated with alter participation. These results suggest that walk-betweeness performs as good as other two traditional centrality measures, or even better considering bigger coefficient and R-squared value, to predict the leaders' influence on the diffusion of microfinance. Building on the village-level regression results of Banerjee et al. (2013), our leader-level analysis suggests that leaders in a central position have direct influence over their followers [56].

When we simultaneously include all three centrality measures in Model 4, only walk-betweeness measure shows statistically significant effects, while the regression coefficients for other measures are significantly attenuated. We further show that the association between walk-betweenness and alter participation is robust against accounting for additional centrality measures such as closeness centrality, betweenness centrality, and eigenvector centrality. Again this result shows that walk-betweenness successfully captures the role of bridging leaders in the diffusion of microfinance.

**Robustness checks.** Conducting robustness checks in Table 6, we show that the effect of walk-betweenness is robust against alternative specifications of network ties and statistical models. In model 1, we use weighted networks instead of binary networks to calculate centrality measures. In model 2, we estimate the effect after excluding followers who were also leaders: a follower is coded as participated only if the follower was not a leader since as the leader could get the information about the microfinance program directly. Our results show that the association between walk-betweenness and alter participation remains similar with our earlier results.

**Table 5. Village-fixed effects regression models for the followers' microfinance take-up rate among leader households in the 75 Indiana villages.**

| Variables | Model 1 | Model 2 | Model 3 | Model 4 | Model 5 |
|---|---|---|---|---|---|
| Walk betweenness (z-score) | 0.17*** | | | 0.19*** | 0.22*** |
| | (0.01) | | | (0.03) | (0.03) |
| Diffusion centrality (z-score) | | 0.14*** | | −0.00 | −0.06 |
| | | (0.02) | | (0.03) | (0.04) |
| Degree centrality (z-score) | | | 0.14*** | −0.03 | −0.12* |
| | | | (0.02) | (0.03) | (0.06) |
| Closeness centrality (z-score) | | | | | −0.02 |
| | | | | | (0.13) |
| Betweenness centrality (z-score) | | | | | 0.06+ |
| | | | | | (0.03) |
| Eigenvector centrality (z-score) | | | | | 0.10+ |
| | | | | | (0.06) |
| Number of rooms | −0.01 | 0.00 | −0.00 | −0.00 | −0.01 |
| | (0.01) | (0.01) | (0.01) | (0.01) | (0.01) |
| Number of beds | −0.02 | −0.02 | −0.02 | −0.02 | −0.02 |
| | (0.02) | (0.02) | (0.02) | (0.02) | (0.02) |
| Number of rooms per capita | −0.03 | −0.06 | −0.04 | −0.03 | −0.03 |
| | (0.06) | (0.07) | (0.07) | (0.06) | (0.06) |
| Number of beds per capita | 0.07 | 0.06 | 0.06 | 0.07 | 0.08 |
| | (0.08) | (0.08) | (0.08) | (0.08) | (0.08) |
| Electricity (ref: Private) | | | | | |
| Government | −0.01 | −0.01 | −0.02 | −0.01 | −0.00 |
| | (0.03) | (0.03) | (0.03) | (0.03) | (0.03) |
| No electricity | 0.04 | 0.02 | 0.03 | 0.04 | 0.05 |
| | (0.05) | (0.05) | (0.05) | (0.04) | (0.05) |
| Latrine (ref: Own) | | | | | |
| Common | 0.08 | 0.06 | 0.05 | 0.08 | 0.09 |
| | (0.10) | (0.13) | (0.11) | (0.10) | (0.10) |
| No latrine | 0.05+ | 0.04 | 0.05 | 0.05+ | 0.05+ |
| | (0.03) | (0.03) | (0.03) | (0.03) | (0.03) |
| Leader's own participation | 0.12*** | 0.12*** | 0.12*** | 0.12*** | 0.12*** |
| | (0.03) | (0.03) | (0.03) | (0.03) | (0.03) |
| Constant | 0.79*** | 0.74*** | 0.78*** | 0.79*** | 0.83*** |
| | (0.04) | (0.05) | (0.05) | (0.05) | (0.08) |
| R-squared | 0.34 | 0.29 | 0.30 | 0.34 | 0.35 |
| Observations | 1,129 | | | | |

Notes. Standard errors in parentheses are clustered at the village level (*** $p<0.001$, ** $p<0.01$, * $p<0.05$, + $p<0.1$). All regressions include fixed effects pertaining to the village.

## Conclusion

We proposed a new measure of centrality named walk-betweenness. Unlike many previous centrality measures that are based on paths, this measure considers the bridging potential of a node (or actor) based on walks that allow any number or repeated lines or nodes in the process of transmission. Walks rather than paths are more appropriate if the transmission (or flow) dynamics are not based on efficiency. We believe many types of transmission are represented more appropriately by walks rather than paths. For example, people do not choose sexual

**Table 6. Village-fixed effects regression models for the followers' microfinance take-up rate among leader households in the 75 Indiana villages: Robust checks.**

| | Model 1 | Model 2 |
|---|---|---|
| **Variables** | **Use of weighted networks** | **Excluding followers who were also leaders** |
| Walk betweenness (z-score) | 0.16*** | 0.23*** |
| | (0.03) | (0.03) |
| Diffusion centrality (z-score) | −0.02 | −0.08* |
| | (0.03) | (0.04) |
| Degree centrality (z-score) | −0.03 | −0.06 |
| | (0.03) | (0.05) |
| Closeness centrality (z-score) | 0.13+ | −0.06 |
| | (0.07) | (0.12) |
| Betweenness centrality (z-score) | 0.02 | 0.02 |
| | (0.02) | (0.02) |
| Eigenvector centrality (z-score) | −0.00 | 0.11+ |
| | (0.03) | (0.06) |
| Number of rooms | −0.00 | −0.02 |
| | (0.01) | (0.01) |
| Number of beds | −0.02 | −0.01 |
| | (0.02) | (0.02) |
| Number of rooms per capita | −0.03 | −0.00 |
| | (0.06) | (0.05) |
| Number of beds per capita | 0.08 | 0.01 |
| | (0.08) | (0.07) |
| Electricity (ref: Private) | | |
| Government | −0.00 | 0.00 |
| | (0.03) | (0.03) |
| No electricity | 0.04 | 0.09+ |
| | (0.04) | (0.05) |
| Latrine (ref: Own) | | |
| Common | 0.08 | 0.15 |
| | (0.09) | (0.10) |
| No latrine | 0.05+ | 0.05 |
| | (0.03) | (0.03) |
| Leader's own participation | 0.12*** | 0.12*** |
| | (0.03) | (0.03) |
| Constant | 0.71*** | 0.82*** |
| | (0.07) | (0.08) |
| R-squared | 0.34 | 0.35 |
| Observations | 1,129 | |

Notes. Standard errors in parentheses are clustered at the village level (*** p<0.001, ** p<0.01, * p<0.05, + p<0.1). All regressions include fixed effects pertaining to the village.

partners to expedite the spread of STDs; that is, STD transmission generally does not follow the shortest paths. Furthermore, if diffusion is examined at the group-level, numerous back-and-forth repeated adoptions and exposure exist between communities, cities, or countries.

The interpretation of walk-betweenness is straightforward. In principle, it indicates the proportion of a transmission that flows through a node in the entire network when repeated exposure through the same node or line are permitted during transmission. We also provided a

transmission correlation matrix to enhance our understanding of the relationship between nodes in the flow process. The correlation ranged from −1 to 1 and was interpreted similar to the typical Fisher correlation coefficient in statistics. This provided another tool to analyze the transmission dynamics by clustering nodes.

We illustrate the potential use of our walk-betweenness measures using aggregated community-level network data for syphilis transmission in the Chicago area. Based on walk-betweenness, we successfully identified hidden bridging communities that were largely ignored because of their low infection rates. Also, our cluster analysis based on the transmission correlation matrix revealed the necessity of joint preventive efforts between geographically distant communities in the same transmission cluster. In addition, we apply our walk betweenness measure to individual-level multilayered network data for the diffusion of microfinance in rural Indian villages. We found that walk-betweenness is as good as other alternative centrality measures when separate estimation was made for each measure. When we simultaneously include all alternative measures in one regression, walk-betweeness outperforms other centrality measures without being attenuated. We believe these results show that walk-betweeness can be more useful to identify central and hidden nodes in various network flows that are based on walks rather than paths or trails.

The identification of a set of nodes with high bridging potential, or say the "influential spreaders", has been also widely studied in the field of optimal percolation on complex networks [59, 60]. For example, Morone and Makse (2015) propose a way to identify the minimal set of nodes to spread of information to the whole network by evaluating their role of maintaining the giant component after being removed based on the conception that the most influential nodes are the ones that enable the whole network to be connected [59]. While our approach differs such measures in optimal percolation given that these measures are based on non-backtracking walks (i.e., a random walk that is not allowed to return back along the edge that it just traversed), future research is needed to compare our measure based on the back-and-forth transmission process against other measures considered in optimal percolation.

Recently, network scholars highlighted the potential of social network analysis as a useful tool to identify the target population in network intervention [61]. For example, during the early stages of epidemics, it is crucial to determine who has the greatest potential for disease transmission [62]. Traditional approaches using degree centrality [63], a simple nomination [64], and more complex measures such as diffusion centrality [56], are all based on the assumption that diffusion processes take place along efficient paths. In this paper, we suggest an alternative and powerful measure—walk-betweenness—to identify the potential target if the diffusion dynamics follow back-and-forth interactions in which walks are the rule rather than the exception.

## Author Contributions

**Conceptualization:** Yoosik Youm.

**Data curation:** Yoosik Youm, Byungkyu Lee, Junsol Kim.

**Formal analysis:** Yoosik Youm, Junsol Kim.

**Funding acquisition:** Yoosik Youm.

**Investigation:** Yoosik Youm.

**Methodology:** Yoosik Youm, Byungkyu Lee, Junsol Kim.

**Project administration:** Yoosik Youm.

**Resources:** Yoosik Youm.

**Software:** Yoosik Youm.

**Supervision:** Yoosik Youm.

**Validation:** Yoosik Youm, Junsol Kim.

**Visualization:** Yoosik Youm.

**Writing – original draft:** Yoosik Youm.

**Writing – review & editing:** Yoosik Youm, Byungkyu Lee, Junsol Kim.

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
