## [Decision Letter · Decision Letter 0]

17 Sep 2020

PONE-D-20-23273

A measure of centrality in cyclic diffusion processes: walk-betweenness

PLOS ONE

Dear Dr. Youm,

Thank you for submitting your manuscript to PLOS ONE. After careful consideration, we feel that it has merit but does not fully meet PLOS ONE’s publication criteria as it currently stands. Therefore, we invite you to submit a revised version of the manuscript that addresses the points raised during the review process.

Please ensure that you address all comments by both reviewers in your revised version.

We look forward to receiving your revised manuscript.

Kind regards,

Federico Botta

Academic Editor

PLOS ONE

Journal Requirements:

2. We note that Figure 5 in your submission contain map images which may be copyrighted. All PLOS content is published under the Creative Commons Attribution License (CC BY 4.0), which means that the manuscript, images, and Supporting Information files will be freely available online, and any third party is permitted to access, download, copy, distribute, and use these materials in any way, even commercially, with proper attribution. For these reasons, we cannot publish previously copyrighted maps or satellite images created using proprietary data, such as Google software (Google Maps, Street View, and Earth). For more information, see our copyright guidelines: http://journals.plos.org/plosone/s/licenses-and-copyright.

2.1.    You may seek permission from the original copyright holder of Figure 5 to publish the content specifically under the CC BY 4.0 license. 

2.2.    If you are unable to obtain permission from the original copyright holder to publish these figures under the CC BY 4.0 license or if the copyright holder’s requirements are incompatible with the CC BY 4.0 license, please either i) remove the figure or ii) supply a replacement figure that complies with the CC BY 4.0 license. Please check copyright information on all replacement figures and update the figure caption with source information. If applicable, please specify in the figure caption text when a figure is similar but not identical to the original image and is therefore for illustrative purposes only.

3. Please include your tables as part of your main manuscript and remove the individual files. Please note that supplementary tables (should remain/ be uploaded) as separate "supporting information" files.

Reviewers' comments:

Reviewer's Responses to Questions

**Comments to the Author**

1. Is the manuscript technically sound, and do the data support the conclusions?

Reviewer #1: Yes

Reviewer #2: Yes

2. Has the statistical analysis been performed appropriately and rigorously? 

Reviewer #1: Yes

Reviewer #2: Yes

3. Have the authors made all data underlying the findings in their manuscript fully available?

Reviewer #1: Yes

Reviewer #2: Yes

4. Is the manuscript presented in an intelligible fashion and written in standard English?

Reviewer #1: Yes

Reviewer #2: Yes

5. Review Comments to the Author

Reviewer #1: The authors introduce a new centrality measure for networks based on walks, rather than paths or trails, to rank nodes accordingly to their 'bridging' potential, i.e., when a node is knocked off the network results dismantled in disconnected components. Walks are indeed more appropriate in describing non-optimised dynamical processes of transport on networks, when many repeated back-and-forth movements qualify diffusive behaviour that are pivotal in many real-world situations, e.g. disease spreading. Although other centrality measures based on walks had already been proposed, e.g. Bonacich's power centrality and Newman's betweenness centrality, the walk-betweeness here introduced is more natural: it does not involve ad-hoc parameters, and it takes into account an effective total count, rather than a net one, of back-and-forth movements. The authors claim that the measure they introduce outperforms other network centrality measures, and they discuss two real-world examples: the spreading of the Syphilis in the Chicago area, and the diffusion of the information subsequent to the introduction of the microfinance in rural areas of India.

Overall the paper is well written, motivated, and easy to understand. In particular, although the introduction may seem a bit lengthy, it gives an in-depth analysis of the state of the art related to the centrality measures which had the greatest impact in sociology studies, and it is pleasant to read. Furthermore, the examples treated are very interesting (I also liked the idea of discussing first an abstract minimal model to make clear the differences with other measures), and I believe the data analysis has been carefully carried out. I only have a few minor comments and suggestions, mainly on the methods and on the choice of a few expressions used by the authors, but I endorse the publication of the paper on PLOS ONE.

Main comments:

- The authors tend to stick to the word 'diffusion' several qualifying adjectives, e.g. efficient, optimal, ecc... I think this may be slightly misleading. Rigorously, in physics, there is no such a thing as an efficient diffusion or an optimal one. Diffusion is diffusion, a very well defined dynamical process (here on networks), characterised by certain well defined features, e.g. the variance grows linearly with time. I think sometimes it would be better to replace the word diffusion with the nouns transport/movement/etc...which, by the way, are better qualified by the adjectives efficient and optimal.

- In 'MATERIALS AND METHODS':

* a Markov chain is a mathematical tool that can be used to model diffusion. Hence, the sentence 'Regular Markov Chain as a Diffusion Process' does not make much sense to me: a Markov chain is not a diffusion process.

* what the author says about the model is fine in my opinion, though they could have slightly simplified the section if they had introduced the weaker concept of ergodic Markov chain (regular implies ergodic). Indeed, to have a stationary probability distribution, it is sufficient to have an ergodic chain.

- In 'Study 1. Diffusion of Syphilis in the Chicago Area': in the abstract and in the introduction the authors claim that the walk betweenness outperforms all the other centrality measures, though in this study no comparison has been made. I see here the necessity to compare the walk-betweenness, at least, with the measures mentioned in the introduction.

Minor comments:

- Pag. 6: in the sentence 'Paths rather than trails or walks are an appropriate diffusion process if the object of diffusion is a valuable resource and if the diffusion does not waste time or energy to repeat nodes or lines' there are two expressions which lead to some confusion, I believe.

Paths/Trails/Walks are not diffusion processes and diffusion is not an agent; I merely intend diffusion as a physical process, and as such it does not decide whether to 'waste' time and energy or not. One can base a centrality measure upon paths/walks/trails, and whether it is going to be a good measure it depends on the nature of the process they want to describe, for instance an optimised transport will agree with Freeman betweenness

centrality and a diffusion with the walk-betweenness.

- pag. 11, first paragraph, end of second sentence: I guess it should be 'bridging potential' and not 'diffusion potential'.

- pag. 11, first paragraph, end of third sentence: typo 'meaninful' -> 'meaningful'.

- pag. 13, line after '[Figure 2 about here]': 'each row of' should be replaced by 'each element of'.

- pag. 14, first line: as it is not revelant for the discussion, I would not mention the word 'equilibrium'. Indeed, even though in mathematics the sense is slightly different, in physics, having a stationary distribution does not imply an equilibrium process, i.e., stationary distribution may also represents out-of-equilibrium processes (think of a ring of states with links allowing only clockwise/anti-clockwise jumps).

- Footnote 7: I believe this is an important part of the methods and should be included in the main text. Furthermore, there is a typo in the first line: 'the mean the first' -> 'the mean first'.

- Both in 'Study 1. Diffusion of Syphilis in the Chicago Area' and 'Study 2. Diffusion of Microfinance in Rural India': when the authors discuss how they handled the data they have, I believe they should be a bit more clear on how for instance they built the necessary matrices (introduced in ' MATERIALS AND METHODS') to perform the data analysis. For instance, it would be helpful to make references with the previous sections.

Ideas and suggestions:

- Ranking nodes according to their bridging potential has been a productive research question even in the field of optimal percolation on networks (see for instance 'Nature 524, 65 (2015)' and 'Proc. Nat. Aca. Sci. 113, 12368 (2016)'). Theoretically speaking it would be nice to see a comparison (not necessarily here in this paper, but maybe in a future work) between walk-betweenness and other measures considered in optimal percolation.

Reviewer #2: The manuscript proposes a new measure of centrality for nodes in complex networks, named “walk betweenness”. This measure tries to assess and quantify the bridging role of each node of a network and is based on walks instead of paths. It is somehow similar to Newman Betweenness, with the difference that it allows back-and-forth diffusion which is instead canceled out by Newman. The Walk Betweenness can provide useful in some specific settings, for instance when the walkers do not have a complete knowledge of the entire network and can only make local choices, or in cases where the diffusing agent follows a walk which is not based on efficiency, like disease of information flow. The authors also show the application to two examples: a Syphilis transmission network and a network of diffusion of Microfinance in India.

The proposed new measure may prove important to analyze specific settings and can be a useful tool to complement the existing centrality measures.

The article is clear and written in good English, the state of the art is well exposed and I suggest its publication on Plos One after some minor changes listed below.

1. The transition matrix T introduced at page 13 is the transition matrix associated to random walk and this should be mentioned. See Noh, Rieger, "Random walks on complex networks." Physical review letters 92.11 (2004): 118701.

2. Page 16: the quantities z and d appearing in the equation are not defined. In general transmission correlation should be better explained, for instance explain the meaning of positive and negative correlation in terms of walk proximity.

3. It would be interesting to see the analogous of fig. 4 appear for other centralities, in particular Newman betweenness.

4. Please provide more descriptive captions for both figures and tables, for instance mentioning which data set they refer to.

5. Please provide a clear reference or repository for Syphilis data at the beginning of the relative section “Data”

6. I suggest an overall second read of the text in order to correct or rephrase some sentences, like for instance “Both data sets are secondary ones that had no personal identifiers from the beginning.”

6. PLOS authors have the option to publish the peer review history of their article (what does this mean?). If published, this will include your full peer review and any attached files.

Reviewer #1: No

Reviewer #2: **Yes: **Giulia Cencetti

---

## [Author Response · Author response to Decision Letter 0]

9 Oct 2020

Reviewer # 1

1) The authors tend to stick to the word 'diffusion' several qualifying adjectives, e.g. efficient, optimal, ... I think this may be slightly misleading. Rigorously, in physics, there is no such a thing as an efficient diffusion or an optimal one. Diffusion is diffusion, a very well defined dynamical process (here on networks), characterised by certain well defined features, e.g. the variance grows linearly with time. I think sometimes it would be better to replace the word diffusion with the nouns transport/movement/etc...which, by the way, are better qualified by the adjectives efficient and optimal.

>>Thanks for your careful reading and suggestion. Sociologists used to use the term, ‘diffusion’, more broadly without exact definition across different contexts; for example, it includes social contagion, disease transmission and the diffusion of ideas, and innovation. That said, we acknowledge that the term could be misleading and as the reviewer suggested, we used the term ‘diffusion’ only when we refer to the general process and use other alternatives terms such as ‘spread’ or ‘transmission’ whenever necessary across the whole manuscript. The following paragraph shows the revision in the introduction section. You can confirm the corrections across the whole manuscript.

We propose a new bridging (brokerage) centrality measure for diverse types of networks that can accommodate an unlimited number of repeated interactions between nodes (actors). Most traditional measures are more appropriately suited for the transmission of valuable goods such as information and assume only optimal and efficient spreading between rational actors; however, our proposed measure is appropriate for disease transmission and web surfing where repeated back-and-forth interaction is the rule rather than the exception. In addition, this measure is well suited for the transmission of information of which the cost is negligible and of which the trustworthiness increases as a result of feedback provided during the back-and-forth transmission process (e.g., the spread of microfinance, fashion, and SNS usage). 

2) a Markov chain is a mathematical tool that can be used to model diffusion. Hence, the sentence 'Regular Markov Chain as a Diffusion Process' does not make much sense to me: a Markov chain is not a diffusion process.

>>The point was fully accepted, and we removed the sub-title, “Regular Markov Chain as a Diffusion Process”. You can confirm this on page 12.

3) what the author says about the model is fine in my opinion, though they could have slightly simplified the section if they had introduced the weaker concept of ergodic Markov chain (regular implies ergodic). Indeed, to have a stationary probability distribution, it is sufficient to have an ergodic chain.

>>As far as we understand, regular chain is an ergodic chain that is not cyclic in mathematical terms. For example, if the transition matrix is . In this case, T is ergodic, but not regular. Since we need a regular not ergodic chain to calculate the walk-betweeness, we would like to stick with the term, “regular” if it is fine with the reviewer. Anyhow, the term appears only twice in the manuscript and thus, if it is still the problem after this re-submission, we will change it without any problem.

4) In 'Study 1. Diffusion of Syphilis in the Chicago Area': in the abstract and in the introduction the authors claim that the walk betweenness outperforms all the other centrality measures, though in this study no comparison has been made. I see here the necessity to compare the walk-betweenness, at least, with the measures mentioned in the introduction.

>>Thanks for your feedback. Note that we have similar comments from Reviewer #2. As Reviewer #2 suggested, we added an analysis based on Newman betweenness score for the comparison: now, figure 4 contains an additional result from the Newman betweeness. As we highlighted in a new manuscript, our walk-betweenness measure reveals more hidden bridging communities than Newman betweenness measure. That said, we admit that it does not provide any conclusive evidence for which measure is better to identify the hidden bridging communities because bridging activity of a node cannot be observed directly. In order to judge which measure performs better empirically, we might need Syphilis incidents data at multiple time points so that we can compare the prediction from walk-betweeness with other alternatives, which is beyond our current study goal. Recall that we emphasized our walk-betweeness that allows multiple, repeated nodes or lines is also conceptually better than traditional centrality measures for the case of disease transmissions. We admit that our original manuscript did not provide enough discussion on this issue and we added more in the revised one. The newly added discussion is as follows.

Also, please note that while we conducted this job, we found a small encoding error in the matrix of communities for Syphilis transmission and corrected the error. As a result, now the figure 4 is a little bit different from the original and, also the cluster analysis produced seven clusters rather than fiver clusters. We double-checked the coding and we are confident that there is no coding error anymore.

That said, Panel B shows that Newman’s betweenness measure reveals fewer hidden bridging communities. This disparity comes from the difference in assumptions between two measures: Newman’s betweeness cancel out back-and-forth opposite transmission through a certain community to calculate net count while walk-betweeness count all the transmissions to obtain total count. We believe that this difference brings a bigger variance to walk-betweeness compared to Neman’s betweeness in general as shown in Figure 4, which could bring more hidden bridges. 

Minor comments:

5) - Pag. 6: in the sentence 'Paths rather than trails or walks are an appropriate diffusion process if the object of diffusion is a valuable resource and if the diffusion does not waste time or energy to repeat nodes or lines' there are two expressions which lead to some confusion, I believe.

Paths/Trails/Walks are not diffusion processes and diffusion is not an agent; I merely intend diffusion as a physical process, and as such it does not decide whether to 'waste' time and energy or not. One can base a centrality measure upon paths/walks/trails, and whether it is going to be a good measure it depends on the nature of the process they want to describe, for instance an optimised transport will agree with Freeman betweenness centrality and a diffusion with the walk-betweenness.

>>Thanks for your comments. We simply agree with this point. Not-so-rare expressions in Sociology, which assumes agents, could be misleading to Physicists. We modified the sentence following Reviewer’s suggestion across the whole manuscript and the following is one example that clarify we assume human agents in the diffusion process. 

If the object of diffusion is a valuable resource and if the agents of the diffusion do not want to waste time or energy to repeat nodes or lines, it would be natural to assume paths rather than trails or walks. Furthermore, shortest paths can be used to examine diffusion processes if the actors have adequate knowledge of the structure of the global network and are able to choose the shortest path. For example, when actors are seeking a piece of imminently necessary information, they prefer to traverse only the shortest paths; thus, people who are occupying bridging positions with high geodesic-betweenness scores obtain influential power. 

6) - pag. 11, first paragraph, end of second sentence: I guess it should be 'bridging potential' and not 'diffusion potential'.

 >> Thanks. We modified the expression following your suggestion.

7) - pag. 11, first paragraph, end of third sentence: typo 'meaninful' -> 'meaningful'.

 >> We corrected the typo as you suggested.

8) - pag. 13, line after '[Figure 2 about here]': 'each row of' should be replaced by 'each element of'.

 >> We corrected the typo as you suggested.

9) - pag. 14, first line: as it is not revelant for the discussion, I would not mention the word 'equilibrium'. Indeed, even though in mathematics the sense is slightly different, in physics, having a stationary distribution does not imply an equilibrium process, i.e., stationary distribution may also represents out-of-equilibrium processes (think of a ring of states with links allowing only clockwise/anti-clockwise jumps).

>> We agree with the point that having a stationary distribution does not imply an equilibrium process. As you suggested, we used the term ‘stationary distribution’ in place of the term ‘equilibrium distribution’.

10) - Footnote 7: I believe this is an important part of the methods and should be included in the main text. Furthermore, there is a typo in the first line: 'the mean the first' -> 'the mean first'.

>> As you suggested, the description in the Footnote 7 was moved to the main text. Also, we corrected the typo as you suggested.

11) - Both in 'Study 1. Diffusion of Syphilis in the Chicago Area' and 'Study 2. Diffusion of Microfinance in Rural India': when the authors discuss how they handled the data they have, I believe they should be a bit more clear on how for instance they built the necessary matrices (introduced in ' MATERIALS AND METHODS') to perform the data analysis. For instance, it would be helpful to make references with the previous sections.

>>We appreciate your suggestion. We added more detailed explanation about network data construction processes under newly added sub-heading, “Network data construction” now. 

Ideas and suggestions:

12) - Ranking nodes according to their bridging potential has been a productive research question even in the field of optimal percolation on networks (see for instance 'Nature 524, 65 (2015)' and 'Proc. Nat. Aca. Sci. 113, 12368 (2016)'). Theoretically speaking it would be nice to see a comparison (not necessarily here in this paper, but maybe in a future work) between walk-betweenness and other measures considered in optimal percolation.

>>Thanks for your insightful suggestion. We agree that it will be important to compare our approach with the existing approaches in the field of optimal percolation on networks although as you said, it is beyond our current study. We, however, discussed this line of studies and pointed out some future directions as follows.

The identification of a set of nodes with high bridging potential, or say the “influential spreaders”, has been also widely studied in the field of optimal percolation on complex networks (Morone and Makse 2015; Braunstein et al 2016). For example, Morone and Makse (2015) propose a way to identify the minimal set of nodes to spread of information to the whole network by evaluating their role of maintaining the giant component after being removed based on the conception that the most influential nodes are the ones that enable the whole network to be connected. While our approach differs such measures in optimal percolation given that these measures are based on non-backtracking walks (i.e., a random walk that is not allowed to return back along the edge that it just traversed), future research is needed to compare our measure based on the back-and-forth transmission process against other measures considered in optimal percolation. 

Reviewer #2

1) The transition matrix T introduced at page 13 is the transition matrix associated to random walk and this should be mentioned. See Noh, Rieger, "Random walks on complex networks." Physical review letters 92.11 (2004): 118701.

>> According to your kind suggestion, we mentioned the random walk and added the reference on page 13 in the revised manuscript.

2) Page 16: the quantities z and d appearing in the equation are not defined. In general transmission correlation should be better explained, for instance explain the meaning of positive and negative correlation in terms of walk proximity.

>> The definition was in footnote 7. We realized that, however, it could be confusing since the definition was hidden in a footnote. Thus, we mentioned this in the main text, now. Also, for a better explanation of the transmission correlation, we added the term, ‘walk proximity’ in two separate occurrences in the main text: page 16 and 20.

3) It would be interesting to see the analogous of fig. 4 appear for other centralities, in particular Newman betweenness.

>>Thanks for your excellent suggestion, and we generated the same figure 4 for Newman betweenness measure. As we discussed briefly, our walk betweenness measure reveals more hidden bridging communities than Newman betweenness measure, which I believe highlights the contribution of our approach.

Also, please note that while we conducted this job, we found a small encoding error in the matrix of communities for Syphilis transmission and corrected the error. As a result, now the figure 4 is a little bit different from the original and, also the cluster analysis produced seven clusters rather than fiver clusters. We double-checked the coding and we are confident that there is no coding error anymore.

4) Please provide more descriptive captions for both figures and tables, for instance mentioning which data set they refer to.

 >>Yes, we provided more descriptive captions for all figures and tables.

5) Please provide a clear reference or repository for Syphilis data at the beginning of the relative section “Data”

 >> According to the suggestion, now we added more reference on page 24.

6) I suggest an overall second read of the text in order to correct or rephrase some sentences, like for instance “Both data sets are secondary ones that had no personal identifiers from the beginning.”

>> Yes, we agree that sentence reads awkward since it was written to show that we used the data without any confidentiality issue following Plos One’s guidelines right before the first submission. Now, we changed the problematic sentence, and improved other sentences as well across the whole manuscript.

---

## [Decision Letter · Decision Letter 1]

27 Nov 2020

PONE-D-20-23273R1

A measure of centrality in cyclic diffusion processes: walk-betweenness

PLOS ONE

Dear Dr. Youm,

Thank you for submitting your manuscript to PLOS ONE. After careful consideration, we feel that it has merit but does not fully meet PLOS ONE’s publication criteria as it currently stands. Therefore, we invite you to submit a revised version of the manuscript that addresses the points raised during the review process.

During review, we were made aware that this work appears similar to a previously published manuscript of which you are an author: 10.1136/sti.2010.044008

In your response to reviewers, please include a section that explains the relationship of this work to the previous manuscript, and specify how this work further advances presentation of the walk betweenness measure. Please clarify any relationship between the datasets used in the two works.

We look forward to receiving your revised manuscript.

Kind regards,

Federico Botta

Academic Editor

PLOS ONE

Reviewers' comments:

Reviewer's Responses to Questions

**Comments to the Author**

1. If the authors have adequately addressed your comments raised in a previous round of review and you feel that this manuscript is now acceptable for publication, you may indicate that here to bypass the “Comments to the Author” section, enter your conflict of interest statement in the “Confidential to Editor” section, and submit your "Accept" recommendation.

Reviewer #1: All comments have been addressed

Reviewer #2: All comments have been addressed

2. Is the manuscript technically sound, and do the data support the conclusions?

Reviewer #1: Yes

Reviewer #2: (No Response)

3. Has the statistical analysis been performed appropriately and rigorously? 

Reviewer #1: Yes

Reviewer #2: (No Response)

4. Have the authors made all data underlying the findings in their manuscript fully available?

Reviewer #1: Yes

Reviewer #2: (No Response)

5. Is the manuscript presented in an intelligible fashion and written in standard English?

Reviewer #1: Yes

Reviewer #2: (No Response)

6. Review Comments to the Author

Reviewer #1: I am satisfied with the way the authors addressed all my previous comments. The manuscript is ready to be published.

Reviewer #2: All comments and suggestions have been addressed, I think that the manuscript is now acceptable for publication.

7. PLOS authors have the option to publish the peer review history of their article (what does this mean?). If published, this will include your full peer review and any attached files.

Reviewer #1: No

Reviewer #2: No

---

## [Author Response · Author response to Decision Letter 1]

30 Nov 2020

Both first-round reviewers were fully satisfied with our resubmitted manuscript and they agreed that the current manuscript is ready to be published.

However, Dr. Federico Botta, Academic Editor requested a section that explains the relationship of this work to one of previous manuscript and specify how this work further advances presentation of the walk betweenness measure. It was a quite valid concern and accordingly, we added a new paragraph to explain the difference between two papers and we are now confident that, it is obvious that our new manuscript advanced the discussion of walk betweeness in many ways: "In your response to reviewers, please include a section that explains the relationship of this work to the previous manuscript, and specify how this work further advances presentation of the walk betweenness measure. Please clarify any relationship between the datasets used in the two works."

Per his request, we added the following paragraph to the manuscript.

 This measure was briefly introduced in a previous study (Youm 2010) without any elaborated mathematical explanation. The paper advances the discussion of walk-betweeness measure in three major ways. First, it provides the full mathematical equations behind the measure and lay out detailed calculations readers need to use to measure the walk-betweeness for themselves. Second, it introduces a new usage of walk-betweeness: a cluster analysis based on transmission coefficients. This cluster analysis can identify non-adjacent or distant neighborhoods who are close to each other in transmission paths. This could be essential to come up with a cooperative preventive strategy of local health-related organizations for transmittable diseases. Finally, it presents empirical tests of walk-betweeness measure by utilizing two new data sets: Syphilis transmission in Chicago area and microfinance diffusion in rural Indian villages.

---

## [Editor Report · Decision Letter 2]

4 Jan 2021

A measure of centrality in cyclic diffusion processes: walk-betweenness

PONE-D-20-23273R2

Dear Dr. Youm,

We’re pleased to inform you that your manuscript has been judged scientifically suitable for publication and will be formally accepted for publication once it meets all outstanding technical requirements.

Kind regards,

Federico Botta

Academic Editor

PLOS ONE
---

## [Editor Report · Acceptance letter]

7 Jan 2021

PONE-D-20-23273R2 

A measure of centrality in cyclic diffusion processes: walk-betweenness 

Dear Dr. Youm:

I'm pleased to inform you that your manuscript has been deemed suitable for publication in PLOS ONE. Congratulations! Your manuscript is now with our production department. 

Kind regards, 

on behalf of

Dr. Federico Botta 

Academic Editor

PLOS ONE